# Canadian Recommendations for Germline Genetic Testing of Patients with Breast Cancer: A Call to Action

**DOI:** 10.3390/curroncol32060290

**Published:** 2025-05-22

**Authors:** Evan Weber, Carlos A. Carmona-Gonzalez, Melanie Boucher, Andrea Eisen, Kara Laing, Jennifer Melvin, Kasmintan A. Schrader, Sandeep Sehdev, Stephanie M. Wong, Karen A. Gelmon

**Affiliations:** 1Division of Medical Genetics, Department of Specialized Medicine, McGill University Health Centre, Montréal, QC H4A 3J1, Canada; 2Department of Human Genetics, McGill University, Montréal, QC H3A 1Y2, Canada; 3Division of Medical Oncology, Odette Cancer Centre, Sunnybrook Health Sciences Centre, Toronto, ON M4N 3M5, Canada; 4Department of Medicine, Division of Medical Oncology, University of Toronto, Toronto, ON M4N 3M5, Canada; 5Department of Oncology, Division of Medical Oncology, PEI Cancer Treatment Centre, Charlottetown, PE C1A 8T5, Canada; 6Department of Oncology, McMaster University, Hamilton, ON L8V 5C2, Canada; 7Cancer Care Program, NL Health Services, St. John’s, NL A1B 3V6, Canada; 8Discipline of Oncology, Faculty of Medicine, Memorial University, St. John’s, NL A1B 3V6, Canada; 9Department of Medicine, Division of Medical Oncology, Nova Scotia Health, Halifax, NS B3H 2Y9, Canada; 10Hereditary Cancer Program, BC Cancer, Vancouver, BC V5Z 4E6, Canada; 11Department of Medical Genetics, University of British Columbia, Vancouver, BC V6T 1Z3, Canada; 12Department of Medicine, Division of Medical Oncology, The Ottawa Hospital Cancer Centre, Ottawa, ON K1H 1C4, Canada; 13Faculty of Medicine, University of Ottawa, Ottawa, ON K1H 8L6, Canada; 14Jewish General Hospital, Stroll Cancer Prevention Centre, McGill University, Montréal, QC H3T 1E2, Canada; 15Gerald Bronfman Department of Oncology, McGill University, Montréal, QC H4A 3T2, Canada; 16Department of Surgery, McGill University, Montréal, QC H3A 3J1, Canada; 17Department of Medical Oncology, BC Cancer, Vancouver, BC V5Z 4E6, Canada; 18Faculty of Medicine, University of British Columbia, Vancouver, BC V6T 1Z3, Canada

**Keywords:** breast cancer, germline genetic testing, Canada, recommendations, mainstreaming, *BRCA1*, *BRCA2*

## Abstract

Pathogenic variants in breast cancer predisposition genes are associated with poor clinical outcomes but also offer an opportunity for more individualized therapeutic pathways. Given increasing knowledge, improvements in germline genetic testing efficiency, and the availability of novel systemic targeted treatment options, the importance of appropriately identifying patients for testing has never been greater. A pan-Canadian expert working group (EWG) consisting of 10 healthcare professionals (HCPs) was convened to review recent international guidelines for germline genetic testing in breast cancer and develop Canadian recommendations. The group identified four clinical questions to address which patients should undergo testing, what approaches should be used, how patients should be counselled, and what steps are needed for implementation. In response to these questions, the EWG agreed upon 12 recommendations that emphasized broader incorporation of germline genetic testing and more standardized, streamlined testing and counselling approaches. The group also offered multiple suggestions to support effective and equitable implementation across Canada. These recommendations provide guidance for HCPs and represent a call to action for the Canadian government and other organizations to support genetic testing pathways, drug access, and ultimately improved outcomes for patients with breast cancer and their families.

## 1. Introduction

Germline genetic testing for breast cancer has evolved considerably over the last 30 years. Identification of pathogenic variants (PVs) in the *BRCA1* and *BRCA2* genes in the 1990s has led to intensified research on molecular pathways of the disease, the development of next-generation sequencing (NGS) and multigene panels, and the introduction of targeted therapeutic options [1,2,3,4,5,6,7]. With approximately 5% to 10% of breast cancers being associated with a germline PV or likely PV (LPV) [8,9,10,11], genetic testing for germline variants is now widely considered an integral component of a comprehensive clinical workup [12,13], with results informing prognosis, hereditary risk, and personalized treatment related to surveillance, risk-reducing measures, and/or systemic agents [14]. These advantages, combined with patent legislation decisions and advances in testing efficiency, have resulted in reduced testing costs, expedited availability of results, and steadily increased demand. To address this growing need, recent efforts have focused on further enhancing testing capabilities, providing timely access to testing through point-of-care consent and test ordering by non-genetics providers (i.e., “mainstreaming”), refining counselling approaches, and implementing tailored management strategies that exploit clinically actionable PVs [15]. These efforts have underscored the importance of careful patient selection for testing to optimize clinical outcomes while maintaining efficiency and avoiding overwhelming healthcare professionals (HCPs) and systems.

In Canada, patients with breast cancer undergo germline genetic testing and receive treatment within a publicly funded healthcare system. Although such systems are generally regarded to have universal implementation, health insurance plans and policies still vary both within and across the country’s vast geography of 13 provinces and territories. Furthermore, no national guidance for germline genetic testing is available for individuals with breast cancer, although independent provincial referral and testing criteria exist [16]. These issues, combined with regional diversity in healthcare organization (e.g., high-volume oncology and genetics services being primarily situated in large centres), population characteristics, and funding priorities, have contributed to variability in access to testing. Nonetheless, although many breast cancer treatment guidelines are also provincially centred, general uniformity is apparent in terms of therapeutic approaches. Therefore, an appetite exists among HCPs for national guidelines for germline genetic testing in this population.

In early 2024, the American Society of Clinical Oncology and the Society of Surgical Oncology (ASCO-SSO) jointly published evidence-based guidelines that specify which patients with breast cancer should be offered germline genetic testing [17]. Recent guidelines presenting similar recommendations, albeit with some variation, are also available from the National Comprehensive Cancer Network^®^ (NCCN^®^) [18], the American Society of Breast Surgeons (ASBrS) [13], and other groups [19,20,21,22]. It was important to understand the applicability of the 2024 ASCO-SSO guidelines and those from other international groups to the Canadian breast cancer setting and to develop recommendations that appropriately acknowledge the complexities of Canada’s publicly funded healthcare system. With this goal in mind, a voluntary pan-Canadian Expert Working Group (EWG) was convened to review current guideline recommendations, identify and deliberate on issues relevant to Canadian patients and HCPs, and prepare tailored recommendations as part of a call to action to promote effective and equitable implementation of best practices for hereditary testing across Canada.

## 2. Materials and Methods

### 2.1. Expert Working Group

In August 2024, a voluntary EWG convened that included ten HCPs: seven medical oncologists (M.B., C.A.C.G., A.E., K.A.G., K.L., J.M., S.S.), one geneticist (K.S.), one surgical oncologist (S.M.W.), and one genetic counsellor (E.W.). These individuals were identified and invited to participate in the EWG based on their expertise in hereditary testing and/or management of breast cancer, as well as their pan-Canadian representation. The size of the EWG was considered sufficient to allow diffuse representation while enabling a cohesive and efficient process.

### 2.2. Preparation, Convergence, and Collaboration

In late September 2024, the EWG members participated in a virtual meeting with the goal of developing Canadian recommendations for germline genetic testing in breast cancer. The 2024 ASCO-SSO guidelines [17] were reviewed, as well as guidelines from other relevant international groups (e.g., ASBrS [13], NCCN Clinical Practice Guidelines in Oncology [NCCN Guidelines^®^] v.3.2025 [18]), Canadian provincial policies, and individual HCP experiences. The EWG specifically considered each ASCO-SSO clinical question and recommendation and discussed pertinent modifications that may better reflect the current Canadian setting.

After the initial meeting, one HCP (K.A.G.) worked with a medical writer to revise the ASCO-SSO clinical questions and recommendations, which were then emailed to the EWG members for feedback. Using an iterative process, two rounds of review and revision of the recommendations were conducted via email and key considerations were documented.

A second virtual meeting was held in early November 2024, during which the EWG deliberated over outstanding issues, finalized and agreed upon the recommendations, and identified additional healthcare system challenges and opportunities meriting acknowledgment from a Canadian standpoint. Overall, differences in opinion were extremely limited; when evident, open discussion occurred until a consensus of opinion was reached. The EWG members self-assigned themselves into smaller cross-specialty groups, each with the responsibility of further discussing and drafting key considerations related to the endorsement or revision of the original ASCO-SSO recommendations.

### 2.3. Reporting of Recommendations

One HCP (K.A.G.) and a medical writer compiled the clinical questions, recommendations, and key considerations provided by each cross-specialty group into one document, and edits were implemented to ensure clarity and to highlight outstanding issues for resolution. The draft manuscript was then shared with the EWG for review and input. In February 2025, a third virtual meeting was held, during which remaining issues were discussed and addressed to ensure consensus. The manuscript was subsequently updated and then distributed to all EWG members for final review and sign-off for journal submission.

## 3. Results and Discussion

Based on review and discussion of the breast cancer germline genetic testing guidelines from ASCO-SSO and other international groups, individual clinical experiences, and needs and challenges specific to the Canadian landscape (e.g., genetic counsellor availability, laboratory result turnaround times, provincial funding models, etc.), the EWG unanimously agreed that Canadianized recommendations would be beneficial. A key objective was to prepare guidance that could equitably, yet efficiently, inform decision-making regarding surveillance, risk-reducing surgical approaches, and systemic treatment options. The EWG also aimed to provide recommendations that address early cancer detection and prevention for both individual patients and their families, including cascade testing and counselling. The group raised numerous pertinent issues and considerations and offered suggestions supporting the implementation of their recommendations across Canada. Critical topics and suggested approaches centred on equity, HCP accessibility, variations in testing pathways (e.g., mainstreaming, coordination, and timing of testing, reporting, and treatment), and drug availability, among others.

It should be noted that the recommendations presented herein reflect a Canadian viewpoint on approaches that are predominantly based on clinical evidence. Given that such evidence is rigorously described elsewhere in previously published guidelines, clinical trial reports, and review articles, a detailed discussion of germline genetic mutations, their epidemiology, and relevant clinical investigations was considered beyond the scope of this initiative. Citations for foundational publications have been included for readers seeking more in-depth information.

### 3.1. Clinical Questions

The experts collectively agreed upon four clinical questions related to germline genetic testing of breast cancer in Canada:Clinical Question 1: Which patients should be offered germline genetic testing for breast cancer?Clinical Question 2: What approaches should be used to offer germline genetic testing for breast cancer, and which genes should be tested?Clinical Question 3: How should patients with breast cancer who are considering genetic testing be counselled in the pre- and post-test setting?Clinical Question 4: What challenges exist and what steps are necessary to implement these recommendations equitably across Canada?

### 3.2. Overview of Recommendations

The EWG prepared and achieved agreement on a total of 12 recommendations regarding approaches to germline genetic testing for Canadian patients with breast cancer (Table 1 and Figure 1). Overall, many of the recommendations remained similar to those published by ASCO-SSO and other groups, although several notable changes were included:Acknowledgement of the clinical value of testing patients with ductal carcinoma in situ (DCIS).Inclusion of concurrent or asynchronous bilateral breast cancers among the testing criteria for patients aged >65 years.Increased emphasis on the importance of breast cancer prevention and early detection in addition to the individualized treatment of affected individuals.Standardized use of multigene panel testing rather than a primary focus on *BRCA1*/*BRCA2*, as well as the specification of additional genes that should be tested at minimum (with the recognition that those recommended may evolve as new evidence becomes available).Increased reflection of Canadian approaches to germline genetic testing in terms of patient flow and non-genetics-provider-initiated testing practices (i.e., mainstreaming).Considerations related to genetic counselling within the Canadian health care system, both in general and with increased use of mainstreaming.Discussion of specific challenges encountered in Canada related to germline genetic testing and guidance on solutions that may support broader and more efficient implementation.
Details regarding these recommendations are provided below.

### 3.3. Recommendations


**Clinical Question 1: Which patients with breast cancer should be offered germline genetic testing?**
**Recommendation 1.1:** All patients with newly diagnosed invasive breast cancer, a personal history of invasive breast cancer, or DCIS who are aged ≤65 years at diagnosis should be offered germline genetic testing.**Recommendation 1.2**: All patients with newly diagnosed invasive breast cancer, a personal history of invasive breast cancer, or DCIS who are aged >65 years at diagnosis should be offered germline genetic testing if one or more of the following criteria are met:

They are candidates for targeted therapies indicated for the presence of germline PVs in early-stage or metastatic disease (e.g., poly[ADP-ribose] polymerase inhibitors [PARPi]).They have triple-negative breast cancer.Their personal or family history suggests the possibility of a PV/LPV (e.g., multiple primary cancers in the individual or family member[s]).They have bilateral breast cancer, either concurrent or asynchronous.They were assigned male sex at birth.They are of Ashkenazi Jewish ancestry or are members of a population with an increased prevalence of founder mutations in relevant genes.

**Recommendation 1.3:** All patients with recurrent breast cancer (local or metastatic) who are candidates for targeted therapies indicated for germline PVs should be offered germline genetic testing.

Germline genetic testing recommendations for breast cancer vary across current international guidelines according to multiple factors, such as breast cancer status (e.g., newly diagnosed, recurrent, personal history), patient age cut-offs, other patient and tumour characteristics, and patient eligibility for targeted therapies. For example, the ASBrS recommends testing all patients with breast cancer [13], whereas ASCO-SSO suggests testing all patients aged ≤65 years and certain patients aged >65 years who present with characteristics recognized to increase risk [17]. The NCCN Guidelines^®^ generally recommend a stricter age cut-off of ≤50 years for testing, although patients of any age may be considered if specified risk criteria or other conditions are met [18]. In Canada, available guidance varies by province/territory without standardization, resulting in differences in approaches across centres and patient inequity in terms of both testing and outcomes. Although recommendations continue to evolve, implementation is frequently delayed given lags in education and uptake (see Clinical Question 4).

Based on current evidence, the EWG recommended that germline genetic testing be offered to all patients who are newly diagnosed with invasive breast cancer or who have a personal history of invasive breast cancer and were diagnosed at ≤65 years of age (Table 1; Figure 1). The selected age cut-off of 65 years is supported by growing research suggesting a clinically relevant presence of germline PVs among female patients aged 50-to-65 years [23,24,25,26,27,28]. For patients with a new diagnosis of invasive breast cancer or a personal history of the disease who were diagnosed at an age >65 years, the EWG recommended that germline genetic testing be offered to individuals meeting certain risk criteria, as per recommendations from ASCO-SSO and NCCN [17,18]. Adding on to criteria related to treatment eligibility, tumour histology, family history, male sex, and the presence of founder mutations, the EWG also specified a criterion for patients presenting with concurrent or asynchronous bilateral breast cancer, flagging that such findings also apply to patients ≤65 years. The group further noted that lesions presenting in different quadrants may represent two primary tumours and are multicentric rather than multifocal disease; as such, decision-making regarding testing should be left to the ordering HCP. In general, the EWG underscored that their recommendations are applicable regardless of tumour size, nodal status, molecular markers, or other pathological findings and that best clinical judgment must be applied to each patient case. As highlighted in the NCCN Guidelines, the EWG also acknowledged that in the absence of recognized risk factors, older individuals (age ≥ 50 years) are more likely to test positive for a PV in a moderate-penetrance gene than in *BRCA1/BRCA2* [18]. The group agreed that the testing of these patients may have more limited clinical utility and that risk management guidelines are often less clear than those for PVs in higher-penetrance genes. Nonetheless, although identification of PVs in moderate-penetrance genes is currently less likely to impact treatment (e.g., surgical decision-making), the EWG expected that affected individuals may still benefit from genetic counselling (see Clinical Question 3).

Unlike other guideline recommendations, the EWG additionally acknowledged the importance of offering germline genetic testing to patients with DCIS. Among female patients with DCIS, the prevalence of germline PVs in *BRCA1/BRCA2* and *PALB2* ranges from 2% to 5% [29,30,31,32]. Furthermore, although studies are limited, individuals aged <50 years diagnosed with in situ malignancy appear to have an increased prevalence of PVs in *BRCA1/BRCA2*, *PALB2*, and *CHEK2*, particularly in the setting of estrogen receptor-positive DCIS [30]. Critically, the detection of PVs in this population can have dramatic implications for local therapy, including consideration of bilateral mastectomy allowing for subsequent omission of adjuvant radiation and/or endocrine therapy. As such, the EWG included patients with DCIS in both Recommendations 1.1 and 1.2, alongside those presenting with invasive breast cancer.

Analogous to other guidelines, the EWG agreed that candidacy for treatment with therapies targeting germline PVs should be a genetic testing criterion for patients aged >65 years with invasive breast cancer (new or personal history) or DCIS, as well as for those with recurrent disease. Presently, such therapy is limited to two PARPi options, olaparib and talazoparib, which are suitable for breast cancer patients of all ages who harbour *BRCA1/BRCA2* PVs and other disease-related characteristics.

The internationally conducted phase 3 OlympiA study compared adjuvant olaparib therapy versus placebo among patients with high-risk early breast cancer that was *BRCA* mutated and human epidermal growth factor receptor 2 (HER2) negative [5,33]. Participants could have either estrogen-negative or estrogen-positive disease. Treatment with olaparib was associated with significantly improved invasive disease-free survival (DFS, the primary study outcome), as well as significant benefits for distant DFS and overall survival (OS). The findings of this landmark trial confirmed the value of *BRCA1* and *BRCA2* as targets in early breast cancer. Olaparib is approved [34] and funded nationally in Canada after receiving a positive health economic recommendation (i.e., reimburse with conditions) from Canada’s Drug Agency (CDA; formerly the Canadian Agency for Drugs and Technologies in Health [CADTH]) for patients with deleterious or suspected deleterious *BRCA*-mutated, HER2-negative breast cancer who meet criteria for high-risk early disease [35], which vary according to estrogen status. The EWG therefore recommended germline genetic testing of all individuals at diagnosis who may benefit from this clinically important treatment option.

Both olaparib and talazoparib have received licensing from Health Canada [34,36] for the treatment of advanced breast cancer in individuals with *BRCA1/BRCA2* PVs based on results of the phase 3 OlympiAD [4,37,38] and EMBRACA [3,39] trials, respectively. However, despite regulatory approval, neither PARPi had received funding for this indication in Canada at the time of this publication—drug manufacturers have not filed reimbursement submissions [40,41], presumably given the non-statistically significant results for OS in the intention-to-treat trial populations [37,38,39]. Regardless of these findings, the therapies are associated with progression-free survival advantages, favourable safety and quality of life outcomes [42,43], and a statistically significant OS benefit among patients who have not already received chemotherapy for metastatic breast cancer (olaparib only; prespecified trial subgroup) [37]. Moreover, open-label, phase 3b evaluation has shown that real-world outcomes with olaparib are comparable to those observed in the OlympiAD study [44]. At present, olaparib is only available to Canadian patients with advanced breast cancer through private insurance coverage, out-of-pocket payment, or manufacturer patient-support programs, while talazoparib is not available for breast cancer therapy in Canada. The EWG unanimously stressed that both olaparib and talazoparib represent clinically important treatment options that should be considered for public funding in the *BRCA*-mutated, HER2-negative setting of advanced breast cancer.

The EWG additionally acknowledged that a preliminary yet growing body of evidence from phase 2 studies, real-world analyses, and case reports indicates that olaparib and talazoparib offer efficacy among patients with advanced breast cancer who harbour germline *PALB2* PVs but no PVs in *BRCA1/BRCA2* [45,46,47,48,49,50]. As the conduct of large studies of patients with *PALB2* PVs is unlikely given the rarity of affected patients, the large number of participants required for a clinical trial, and the need for significant study funding, new evidence may not be forthcoming. Regardless, based on current findings and the mechanism of *PALB2*, the EWG supported the consideration of olaparib in *PALB2*-mutated advanced breast cancer, given its potential benefits. Reviewing the evidence collectively, the EWG strongly encouraged both manufacturers and reimbursement bodies to increase access to PARPis to improve outcomes for patients with advanced disease, who otherwise suffer from a limited number of treatment options.


**Clinical Question 2: What approaches should be used to offer germline genetic testing for breast cancer, and which genes should be tested?**
**Recommendation 2.1:** Germline genetic testing should be included in the initial assessment of patients via mainstreaming or other modalities that ensure a timely and efficient approach.**Recommendation 2.2:** Germline genetic testing should use next-generation sequencing with a multigene panel that includes, but is not limited to, the following genes: *ATM*, *BARD1*, *BRCA1*, *BRCA2*, *CDH1*, *CHEK2*, *NF1*, *PALB2*, *PTEN*, *RAD51C*, *RAD51D*, *STK11*, and *TP53*.

Recent developments in genetic testing have decreased costs and accelerated the identification of a broader range of clinically significant genetic abnormalities, thereby facilitating more personalized management of breast cancer and other malignancies [9]. Given this, the EWG agreed that for the vast majority of patients, germline genetic testing can and should occur reflexively at breast cancer diagnosis such that treatment decision-making can be informed as quickly as possible, as well as judgments regarding cascade testing. The group stated that in Canada, mainstreaming and similar streamlined approaches represent effective and collaborative modalities for rapid and equitable testing and determination of next steps [19,51,52,53,54]. Typically initiated by the treating medical or surgical oncologist, the mainstreaming pathway includes discussing genetic testing and its potential implications with the patient, requesting testing and reviewing results, and ensuring referral to clinical genetics as appropriate [19,51]. Through this process, mainstreaming reduces the need for specialized guidance from genetic counsellors, who remain a limited resource in Canada [55]. As only about 5% of unselected breast cancer cases are associated with a germline PV in a breast cancer susceptibility gene [8,9,10,11,56], many patients do not require the individualized pre-test genetic counselling provided by clinical genetics services; for those who do (see Clinical Question 3), mainstreaming allows genetic counsellors to better focus their efforts and skillset on complex cases or persons being seen for cascade testing. By streamlining genetic testing processes, mainstreaming can expedite access to PV results and quickly enable the initiation of more individualized therapy and management among both patients with breast cancer and blood-related family members at risk of carrying familial PV/LPV (see Clinical Question 3). Still, as discussed in Clinical Question 4, the EWG recognized that such pathways vary across centres and provinces/territories and that bottlenecks exist related to education, staffing, and funding.

In Canada and globally, NGS and multigene panels are the standard of care for assessing germline genetic mutations among patients with breast cancer. Advocated by ASCO-SSO and others, expanded-panel testing enables the identification of clinically actionable PVs beyond *BRCA1/BRCA2* and may inform the need for assessment of second primary cancers and familial risk [17,57,58]. Currently, the specific genes included in these panels vary on the basis of local guidelines, clinical and laboratory resources, HCP education, patient characteristics, and/or the availability of genetic expertise [1,17]. Given that cancer prevention and treatment are of similarly high importance, the EWG felt it was imperative to specify a minimum panel of genes that should be evaluated among patients with breast cancer. In accordance with recent guidelines from the European Molecular Genetics Quality Network (EMQN), the group agreed that gene selection should be informed by clinical utility [19]; therefore, taking into consideration current clinicogenetic evidence, recommendations from other professional organizations and commercially available gene panels (including those summarized recently by Menko and colleagues [1]), and the characteristics of the Canadian population, the EWG specified 13 genes that should be assessed using NGS techniques. At minimum, panels should include the high-penetrance genes *BRCA1*, *BRCA2*, and *PALB2*, as well as several moderate-penetrance and/or syndromic genes: *ATM*, *BARD1*, *CDH1*, *CHEK2*, *NF1*, *PTEN*, *RAD51C*, *RAD51D*, *STK11*, and *TP53*. These selections were prioritized by the EWG given that PVs in these genes are associated with a relative risk ≥2.0 for breast cancer, a risk of a more aggressive breast cancer subtype (e.g., triple-negative disease), and/or elevated non-breast cancer risks (e.g., other cancer types) [59,60]. The EWG acknowledged that although germline PVs in moderate-penetrance genes do not typically confer a significant risk of contralateral breast cancer such that they would be used in surgical planning, exceptions exist (e.g., *ATM* c.7271T>G) and additional data may soon be available.


**Clinical Question 3: How should patients with breast cancer who are considering germline genetic testing be counselled in the pre- and post-test setting?**
**Recommendation 3.1:** All patients who are candidates for germline genetic testing should be given sufficient information before testing to support their informed consent.**Recommendation 3.2:** All patients with a PV/LPV should be provided individualized post-test genetic counselling and offered a referral to a provider experienced in clinical cancer genetics.**Recommendation 3.3:** Identification of a VUS typically should not alter management. Patients should be made aware that although most VUS are eventually reclassified as non-disease causing or benign, such variants may occasionally be reclassified as pathogenic. As VUS reassessment and reporting of reclassification are not standardized across laboratories, ordering providers should be aware of their local testing laboratory’s practices. Consultation with a provider experienced in clinical cancer genetics can be helpful and should be made available, especially if a patient’s personal or family history is suspicious for a hereditary cancer syndrome or they have ongoing concerns regarding the impact of a VUS despite explanation.**Recommendation 3.4:** Patients without a PV on germline genetic testing may benefit from counselling if there is a significant personal or family history of cancer. Referral to a provider experienced in clinical cancer genetics is especially recommended if the patient’s personal or family history is suspicious for a hereditary cancer syndrome, regardless of the patient’s negative test result. Consultation between the HCP and the cancer genetics service and referral to a provider experienced in clinical cancer genetics can be helpful when there is uncertainty.

In alignment with ASCO-SSO, the EWG flagged the importance of patients receiving clear, comprehensive, and personalized information supporting decision-making as part of the mainstreaming process. Such information should provide a general understanding of hereditary cancer, describe the benefits and potential risks of genetic testing, and set expectations regarding the testing process, possible outcomes, the impact on prevention and treatment selection, and implications for family members [61,62,63,64]. Ideally, this information should be made available in multiple formats (e.g., websites, videos, brochures, one-on-one and group discussions) to support variations in learning styles, as well as in several languages given the cultural diversity of the Canadian population. The use of pre-test counselling checklists can promote standardized provision of genetic testing information across HCPs, centres, and provinces [64]. The EWG emphasized that in addition to informing patients, it is equally critical that HCPs receive ongoing training and education to effectively communicate complex genetic concepts and results. These professionals must be equipped to interpret genetic findings accurately, provide emotional support, and discuss potential next steps in a way that is tailored to each patient’s unique context and preferences. This combination of patient education and HCP training ensures that patients are empowered to make decisions that align with their values, while also fostering trust in the healthcare system—when both patients and HCPs are adequately prepared, genetic testing offers a valuable tool in personalized breast cancer care.

Parallel to ASCO-SSO, the EWG agreed that referral to clinical genetics should occur for all patients who test positive for a PV/LPV in a breast cancer susceptibility gene. The group stressed that among the different patient groups, these individuals represent the most important cohort requiring referral: consultation can provide information or counselling that may lead to modification of the treatment plan, guide recommendations for screening and/or risk reduction for new primary cancers (e.g., ovarian cancer), as well as inform cascade testing of family members with the goal of early cancer detection or prevention. As evidence for the clinical impact of PVs continues to evolve, and as many cases remain associated with insufficient clarity regarding optimal management approaches [17,57], affected patients may also be candidates for participation in ongoing research that will increase understanding of their particular PV(s) and clinical outcomes.

The EWG appreciated that the identification of a VUS may raise significant concerns, as these variants are often poorly understood by both HCPs and patients. They noted that although VUS identification does not typically change the treatment plan, clinical and cancer genetics teams must proactively collaborate to manage affected patients appropriately. The EWG stated that for patients with a VUS or who have a strong family history but no PV/LPV, a referral should be made to the specialized genetics program as it may lead to expanded genetic testing, additional investigation to better clarify the VUS, or tailored cancer screening and risk-reduction recommendations. Moreover, cancer genetics service registration may allow these individuals to benefit from future testing opportunities and clinical research studies. The EWG acknowledged that it is more common for a VUS to be downgraded to likely benign/benign than elevated to a PV/LPV; regardless, they underscored that given variations in practice, ordering HCPs must be aware of their local laboratory’s protocols related to sharing information on VUS reclassification. Furthermore, patients must be apprised of their personal follow-up responsibilities, such as periodically re-contacting the ordering HCP to inquire about updates to VUS classification. The EWG anticipated that as oncology teams gain more experience with mainstream testing, their ability to explain VUS will improve and fewer patients may require referral to genetics groups. Similarly, as databases continue to expand and include more individuals from around the world—especially those of non-European origin—knowledge surrounding VUS incidence will increase and provide more reassuring data regarding the interpretation of such variants.


**Clinical Question 4: What challenges exist and what steps are necessary to implement these recommendations equitably across Canada?**
**Recommendation 4.1:** Policy changes and frameworks are needed to support expanded education, testing, counselling, and clinical follow-up needs relating to germline genetic testing of breast cancer in Canada.**Recommendation 4.2:** A national guideline for genetic testing should be developed to improve consistency and uptake across the country and to provide a foundation for funding in each province/territory.**Recommendation 4.3:** A national working group of experts in cancer and genetics should be established to provide expertise to the provinces/territories and to ensure national standards are communicated and maintained as genetic testing evolves.

In Canada and elsewhere, broader implementation of germline genetic testing recommendations for patients with breast cancer is limited by logistical, financial, and ethical challenges. These obstacles restrict equitable access to testing, impair patients’ ability to make immediate decisions regarding treatment, and limit opportunities for cascade testing that could prevent many future cancers.

### 3.4. Key Challenges

Although cost and accessibility remain major hurdles to germline genetic testing in Canada, it is important that HCPs and payers recognize the clinical and economic value of testing [61,62,63,64]. Limited awareness of and education on germline genetic testing among both HCPs and patients are also persistent barriers. Healthcare professionals may feel inadequately equipped, with challenges related to test ordering and interpretation impeding confidence or interest in participating in mainstreaming approaches [65] and leading to limited or inconsistent patient referrals. Furthermore, patients with breast cancer may not fully understand the value of genetic testing, and concerns surrounding privacy and insurance implications may result in fear and hesitation. Infrastructure and resource limitations, such as an ongoing shortage of genetic counsellors in the face of increasing demand, can further exacerbate these issues and prolong wait times. Faster standardized turnaround times are needed for genetic testing, particularly for patients for whom findings may inform immediate decision-making across diverse treatment options. High-volume cancer centres and medical genetics clinics primarily being located within larger cities also represents a challenge: given Canada’s vast geography, a substantial proportion of the population lives far away from these resources, and thus breast cancer patients residing in rural areas may have historically had limited access to genetic services. Minimization of disparities in education and testing practices between rural and urban centres must continue to be a priority.

### 3.5. Proposed Solutions

The EWG underscored that a comprehensive and efficient national program is required to better support germline genetic testing of breast cancer. At a minimum, policy changes are needed to support expanded education, testing, counselling, and clinical follow-up needs. Funding policies must be reviewed and evolve to support mainstream testing and timely access, as some provinces/territories currently mandate genetic counsellor assessment of every cancer genetics case. Ongoing provision of education and training of HCPs is essential and public health campaigns aimed at patients and family doctors can increase awareness. Uncomplicated patient information (see Clinical Question 3) and consent processes should be applied. Robust legal frameworks protecting against genetic discrimination must continue and be promoted to alleviate ethical concerns—information regarding the protections for Canadians, afforded through the Genetic Non-Discrimination Act around genetic discrimination, should be highlighted [66]. Cancer registry data can be mined to identify patients eligible for testing, particularly for the prevalent population for whom testing may be rolled out gradually. Primary care practitioners can help manage mainstreaming processes for historically prevalent patients who meet recommended criteria. Institutions and organizations, such as the Canadian Partnership Against Cancer (CPAC) [67], the Canadian Task Force on Preventive Health Care [68], and centres with a specialized interest in genetics (e.g., Women’s College Hospital), as well as advocacy groups such as the Canadian Breast Cancer Network, Breast Cancer Canada, and Rethink, could play a leading role in outlining an implementation plan using a public health approach.

As highlighted throughout their recommendations, the EWG agreed that a more uniform and broader approach to testing will help simplify algorithms and eligibility review processes. Synoptic pathology reports can be tailored to remind clinicians of testing options after breast cancer diagnosis. In settings where sensitivity for detection of all clinically actionable germline variants is available via tumour sequencing, there can be consideration of further refinement of reflexive testing processes through the use of tumour-first approaches; these have already been successfully implemented in ovarian cancer [69]. For all patients in whom tumour testing identifies a PV in a breast cancer predisposition gene, germline genetic testing should be undertaken to confirm the presence of the variant(s). At this time, the presence of a somatic mutation does not generally affect breast cancer treatment, despite data showing the efficacy of PARPi [44,70]; however, practices may change in the future as evidence and therapeutic options continue to evolve. Similarly, other inherited genes can have an impact on homologous recombination deficiency (HRD)-related breast cancer [71]. If new therapeutic agents are able to successfully target HRD-related disease, identification of relevant gene variants (e.g., outside *BRCA*, *PALB2*, etc.) will be important for treatment decision-making.

The EWG suggested that to expedite the delivery of genetic testing results, a standardized hierarchical framework could be implemented to provide more optimal and pragmatic care. For example, turnaround targets could be set at one-to-two weeks for patients who require an immediate surgical decision, two-to-four weeks for individuals eligible for PARPi therapy in the adjuvant or advanced disease setting, or four-to-eight weeks to support proper planning for patients with early breast cancer who initiate neoadjuvant therapy. In the settings of screening and cascade testing of unaffected family members, longer wait times may be acceptable; nonetheless, a general turnaround time of less than 12 weeks is likely optimal, given longer timing may prompt unnecessary anxiety.

The EWG additionally flagged the need for increased integration of genetic services into routine cancer care activities via mainstreaming procedures to further support streamlined access. The inclusion of more comprehensive patient information (e.g., family history) routinely on testing requisitions could better support genetics teams in interpreting variants. The use of telehealth could support expanded access to genetic counselling for patients with PVs and VUS, as well as improve guidance related to cascade testing. Novel digital solutions may also help better identify target patients, speed the dissemination of results, and support patients located in rural locations or with special needs. In Canada, healthcare in remote areas is usually incorporated into larger urban cancer centres through a robust system of virtual care, as well as local general practitioners with a focus on oncology. The EWG acknowledged that this system was strengthened in recent years because of needs associated with the COVID-19 pandemic. They also noted that although genetic services have similarly expanded in this fashion, further enhancement of rural physician education and telemedicine is needed to ensure equitable services are available to all Canadians. With genetic testing being incorporated into oncology care, current disparities may be easier to overcome.

Finally, the EWG emphasized that as an increasing number of patients are identified to have PVs, with or without a history of cancer, access to downstream resources, such as breast magnetic resonance imaging, risk-reducing surgery, and monitoring within high-risk clinics, must be guaranteed. Furthermore, as highlighted above, given the demonstrated clinical benefits, PARPi therapy should be made available to all eligible patients. By addressing these challenges, broader genetic testing of breast cancer patients can be more effectively implemented to improve early detection, administration of personalized treatments, and management of familial risk.

## 4. Conclusions

Current evidence strongly supports the importance of reflexive, widespread germline genetic testing to facilitate individualized treatment of breast cancer and prevent or catch early, new malignancies in patients and their family members. The recent evolution of therapeutic options has further increased the urgency to identify patients who may benefit from precision medicine. Tailored follow-up and surgical approaches have been employed for many years; however, the striking clinical benefits associated with adjuvant olaparib therapy in early breast cancer, including improved OS, have further pushed HCPs to recognize PV identification as an integral component of a comprehensive breast cancer work up. As recent epidemiological data suggest an increasing breast cancer incidence [72,73,74], particularly among younger individuals, there is a substantial need to more effectively identify individuals at elevated risk.

The recommendations presented herein adapt current international guidelines to the Canadian setting of germline genetic testing with the aim of supporting optimal and equitable care of patients with breast cancer and their families. This guidance highlights specific patient groups (e.g., DCIS, concurrent or asynchronous bilateral breast cancer) and considerations (e.g., standardized minimum multigene panel, expanded mainstreaming pathways, broader access to PARPi therapy) that have not been called out explicitly in the key recommendations published by other groups. The authors appreciate that some Canadian centres and HCPs may be overwhelmed with the concept of enhanced testing recommendations; nonetheless, it is essential that pathways are identified to adopt this guidance quickly so that patient outcomes can be improved and new cancers can be avoided. The authors also recognize that other jurisdictions may face different challenges related to medical resources, genetic profiles, and cultural differences. However, this guidance may offer a framework that can be tailored regionally. Ultimately, these recommendations represent a call to action for governments and HCPs to institute best practices that will decrease unnecessary mortality and suffering from breast cancer and other hereditary cancers. We call on our Canadian leaders to act with us.

## Figures and Tables

**Figure 1 curroncol-32-00290-f001:**
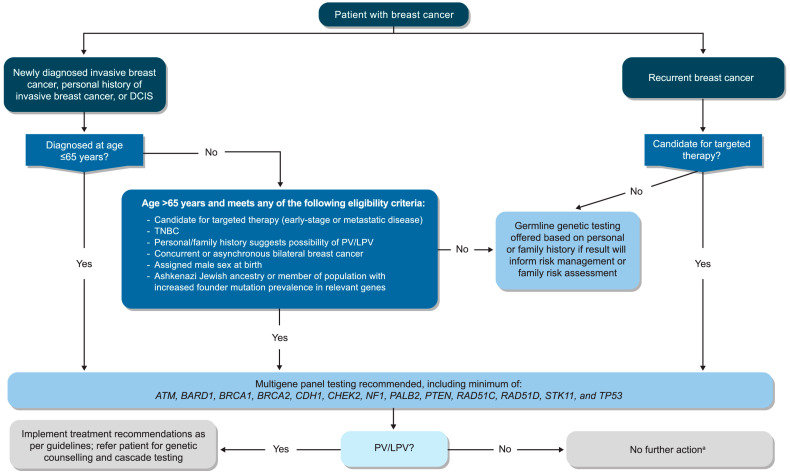
Algorithm for germline genetic testing of patients with breast cancer in Canada. ^a^ If a patient is negative for a PV/LPV but has a significant family history or a VUS, referral to a specialized genetics program may be appropriate, as it may lead to expanded genetic testing, additional investigation, or tailored cancer screening and risk-reduction approaches; see Recommendations 3.3 and 3.4. If a patient requests additional information or could benefit from a more in-depth discussion of their VUS, referral to a genetic counsellor is likely appropriate. All patients with a VUS should be apprised of their personal follow-up responsibilities given the potential for re-classification. DCIS, ductal carcinoma in situ; LPV, likely pathogenic variant; PV, pathogenic variant; TNBC, triple-negative breast cancer; VUS, variant of uncertain significance.

**Table 1 curroncol-32-00290-t001:** Summary of Canadian recommendations for germline genetic testing of breast cancer.

**Guideline Question**	Which Canadian patients should be offered germline genetic testing for PVs in breast cancer susceptibility genes?
**Target Audience**	Medical oncologists, radiation oncologists, surgical oncologists, medical geneticists, genetic counsellors, oncology nurses, oncology advanced practice providers, primary care practitioners, patients, caregivers
**Clinical Question 1: Which patients with breast cancer should be offered germline genetic testing?**	
**Recommendation 1.1:** All patients with newly diagnosed invasive breast cancer, a personal history of invasive breast cancer, or DCIS who are aged ≤65 years at diagnosis should be offered germline genetic testing. **Recommendation 1.2:** All patients with newly diagnosed invasive breast cancer, a personal history of invasive breast cancer, or DCIS who are aged >65 years at diagnosis should be offered germline genetic testing if one or more of the following criteria are met: They are candidates for targeted therapies indicated for the presence of germline PVs in early-stage or metastatic disease (e.g., PARPi). They have triple-negative breast cancer. Their personal or family history suggests the possibility of a PV/LPV (e.g., multiple primary cancers in the individual or family member[s]). They have bilateral breast cancer, either concurrent or asynchronous. They were assigned male sex at birth. They are of Ashkenazi Jewish ancestry or are members of a population with an increased prevalence of founder mutations in relevant genes. **Recommendation 1.3:** All patients with recurrent breast cancer (local or metastatic) who are candidates for targeted therapies indicated for germline PVs should be offered germline genetic testing.	
**Clinical Question 2: What approaches should be used to offer germline genetic testing for breast cancer, and which genes should be tested?**	
**Recommendation 2.1:** Germline genetic testing should be included in the initial assessment of patients via mainstreaming or other modalities that ensure a timely and efficient approach. **Recommendation 2.2:** Germline genetic testing should use next-generation sequencing with a multigene panel that includes, but is not limited to, the following genes: *ATM*, *BARD1*, *BRCA1*, *BRCA2*, *CDH1*, *CHEK2*, *NF1*, *PALB2*, *PTEN*, *RAD51C*, *RAD51D*, *STK11*, and *TP53.*	
**Clinical Question 3: How should patients with breast cancer who are considering germline genetic testing be counselled in the pre- and post-test setting?**	
**Recommendation 3.1:** All patients who are candidates for germline genetic testing should be given sufficient information before testing to support their informed consent. **Recommendation 3.2:** All patients with a PV/LPV should be provided individualized post-test genetic counselling and offered a referral to a provider experienced in clinical cancer genetics. **Recommendation 3.3:** Identification of a VUS typically should not alter management. Patients should be made aware that although most VUS are eventually reclassified as non-disease causing or benign, such variants may occasionally be reclassified as pathogenic. As VUS reassessment and reporting of reclassifications are not standardized across laboratories, ordering providers should be aware of their local testing laboratory’s practices. Consultation with a provider experienced in clinical cancer genetics can be helpful and should be made available, especially if a patient’s personal or family history is suspicious for a hereditary cancer syndrome or they have ongoing concerns regarding the impact of a VUS despite explanation. **Recommendation 3.4:** Patients without a PV on germline genetic testing may still benefit from counselling if there is a significant personal or family history of cancer. Referral to a provider experienced in clinical cancer genetics is especially recommended if the patient’s personal or family history is suspicious for a hereditary cancer syndrome, regardless of the patient’s negative test result. Consultation between the HCP and the cancer genetics service and referral to a provider experienced in clinical cancer genetics can be helpful when there is uncertainty.	
**Clinical Question 4: What challenges exist and what steps are necessary to implement these recommendations equitably across Canada?**	
**Recommendation 4.1:** Policy changes and frameworks are needed to support expanded education, testing, counselling, and clinical follow-up needs relating to germline genetic testing of breast cancer in Canada. **Recommendation 4.2:** A national guideline for genetic testing should be developed to improve consistency and uptake across the country and to provide a foundation for funding in each province/territory. **Recommendation 4.3:** A national working group of experts in breast cancer and genetics should be established to provide ongoing expertise to the provinces/territories and to ensure national standards are communicated and maintained as genetic testing evolves.	

DCIS, ductal carcinoma in situ; EWG, Expert Working Group; HCP, healthcare professional; LPV, likely PV; PARPi, poly(ADP-ribose) polymerase inhibitor; PV, pathogenic variant; VUS, variant of uncertain significance.

## Data Availability

Data are contained within the article.

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
