# Peer review of "Canadian Recommendations for Germline Genetic Testing of Patients with Breast Cancer: A Call to Action"

_curroncol, 2025, doi:10.3390/curroncol32060290_

Round 1
Reviewer 1 Report
Comments and Suggestions for Authors
Thank you for the chance to review your paper. I have provided some big picture and then some more specific suggestions to improve the paper below.
Big picture:
*Consider adding another paragraph to better "set the stage" for the current state in Canada in the introduction. This will be especially important for readers who are not from Canada and don't know the setting as well. Some of the information that comes in the "key challenges and proposed solutions" section, specifically, might be helpful up front.
*Similarly, consider starting the paper with a strong statement of the problem you are addressing (consider lines 77-80). I didn't understand the implications/importance of national guidelines and between location variability until later in my reading.
*Give readers more of a grounding in how national guidelines in cancer work in Canada to better understand if this is a broader problem (that germline testing is one instance of) or unique to this clinical area, specifically.
*In the conclusions, highlight the unique and impactful statements that this group makes: e.g., how they differ w/ respect to DCIS, need for coverage of various therapies.
Specific suggestions:
*line 98-why potential goal and not just goal?
*section 2.3- this section seemed odd to me, that the primary reporting output was a peer reviewed publication vs something with more of a direct way to lead to guidelines. Do you want to highlight the development of this paper specifically? Are there other types of reporting and dissemination you did or plan to do that you can add to this section?
*Table 1 is too big to be of use. Also the into material is confusing and I think not needed (target population, methods). One suggestion is to break this table up into 4 tables (one for each question) and have those Tables lead off each section instead of a restating of the recommendations in text.
*Define (briefly) mainstreaming earlier in the paper.
*Line 315- why is a study w/ PALB2 not feasible?
*some sections have subheadings and others don't; consider standardizing
*Line 560- consider replacing unequivocally with strongly
*Line 570- provide a citation
Reviewer 2 Report
Comments and Suggestions for Authors
This article was about the Canadian recommendations for the germline testing in patients with breast cancer and its significant practical implications were obvious. In response to the current situation that Canada lacks a national guideline for germline genetic testing of breast cancer, this article established a working group composed of experts from multiple fields, aiming to develop testing recommendations suitable for the national conditions. By comprehensively considering international guidelines, provincial policies, and clinical experiences, the article proposed a series of recommendations covering testing targets, methods, counseling, and implementation steps. However, it also has certain limitations.
- In terms of research methods, the expert team of only 10 members is relatively small, which may not be able to fully represent the complex and diverse medical environment and patient groups in Canada, resulting in insufficient representativeness.
- This article does not clearly explain how to resolve differences among experts, affecting the scientific nature and consistency of the recommendations.
- Regarding research data, detailed information about data sources, such as sample size and geographical distribution, is not clearly mentioned.
- For remote areas or special populations in Canada, the feasibility and effectiveness of the recommendations also need to be further verified.
- The recommendations are mainly developed for the Canadian medical system and patients. When applied in other countries or regions, they may be restricted by factors such as medical resources, genetic backgrounds, and cultural differences. I would suggest adding some discussion about the generalizability of the conclusions.
Reviewer 3 Report
Comments and Suggestions for Authors The context is emphasized, with consideration of provincial differences, funding issues, and health care infrastructure. Committed to evidence-based medicine 1. If you have the data, slightly expand the reach of health services and underserved populations. 2. Consider providing a decision support tool or framework for clinicians in managing VUS (variants of uncertain significance) in complex family histories.Author Response
Please see the attachment.
